# Investigating the Dynamic Variation of Skin Microbiota and Metabolites in Bats During Hibernation

**DOI:** 10.3390/biology14121648

**Published:** 2025-11-23

**Authors:** Fan Wang, Wendi Song, Denghui Wang, Zihao Huang, Mingqi Shan, Shaopeng Sun, Zhouyu Jin, Jiaqi Lu, Yantong Ji, Keping Sun, Zhongle Li

**Affiliations:** 1College of Life Science, Jilin Agricultural University, Changchun 130118, China; 2Jilin Provincial Key Laboratory of Animal Resource Conservation and Utilization, Northeast Normal University, Changchun 130117, China; 3Jilin Provincial International Cooperation Key Laboratory for Biological Control of Agricultural Pests, Changchun 130118, China

**Keywords:** *Pseudogymnoascus destructans*, *Rhinolophus ferrumequinum*, skin, microbiome, metabolites

## Abstract

The microbial communities and metabolites on animal skin play a vital role in regulating host immunity and defending against diseases. In bats, skin-associated microbes and metabolites have been shown to inhibit the growth of the fungal pathogen (*Pseudogymnoascus destructans*). However, knowledge about its dynamic changes during hibernation remains limited. Therefore, in this study, we used greater horseshoe bats (*Rhinolophus ferrumequinum*) as a model to investigate these dynamics from hibernation by integrating high-throughput sequencing with untargeted metabolomics. This approach revealed temporal variations in skin bacterial communities and metabolites and elucidated their potential interactions in pathogen defense. Our findings provide important insights into the relationships among microbiota, metabolites, and pathogens, offering a theoretical foundation for the prevention and control of fungal diseases in bats and for developing microbial-based intervention strategies in wildlife.

## 1. Introduction

Bats (*Chiroptera*) are the only mammals capable of sustained flight, occupying a unique ecological niche in nocturnal environments. They play a vital ecological role as major predators of agricultural pests and as pollinators of numerous plant species [1]. However, the emergence of fungal infectious diseases has posed a serious threat to global bat populations. Among these, white-nose syndrome (WNS), caused by *Pseudogymnoascus destructans* (*Pd*), has resulted in the deaths of millions of hibernating bats across North America, with some populations facing the risk of local extinction [2]. Notably, the intensity of fungal infection varies substantially among bat species. Compared with their North American species, bats exhibit considerably lower infection levels in China, with fungal loads only about one-thousandth of those observed in highly susceptible species [3]. This suggests possible co-evolution with the pathogen or higher host resistance, among other environmental factors.

The skin is the largest organ in mammals and is colonized by a complex microbial community comprising bacteria, fungi, and viruses [4]. The skin microbiome can adapt rapidly to environmental fluctuations and pathogen pressures, thereby helping maintain host health [5]. Previous studies have shown that the composition of bat skin microbiome is shaped by multiple factors—including environmental conditions, host physiology, and pathogen infection—and exhibits dynamic variations across temporal and spatial scales [6,7]. Current research on bat skin microbiota has mainly focused on community composition, structure, function, and influencing factors, as well as on the ecological processes driving microbiome variation and the isolation and identification of antimicrobial microorganisms [8,9].

Metabolomics enables the comprehensive characterization of an organism’s metabolic state, reflecting the metabolic features of both the host and its microbiome under physiological and pathological conditions. The composition of skin metabolites can be influenced by multiple factors, including skin cell and glandular secretions, the resident microbiome, and external environmental conditions [10,11,12]. For instance, with advancing age, the homeostasis of hydrocortisone changes, and the metabolic level of retinoic acid decreases [13]. Compared with lowland regions, facial skin in northwestern areas shows lower levels of eicosanoids and total lipids [14]. Moreover, studies have shown that the concentrations of intestinal metabolites sphingosine-1-phosphate and taurine increase with altitude. Notably, sphingosine-1-phosphate exhibits a significant positive correlation with 58 microbial functional modules, including those involved in fatty acid biosynthesis [15]. Identifying metabolites that are produced or modified by the microbiota provides valuable insight into microbial regulatory mechanisms and their influence on host metabolic processes. For example, phenazine-1-carboxylic acid and volatile organic compounds produced by *Pseudomonas yamanorum* GZD14026 isolated from bat skin effectively inhibit *Pd* growth at low concentrations [16]. Under natural conditions, the skin commensal bacterium (*Janthinobacterium lividum*) in *Plethodon cinereus* produces high concentrations of indole-3-carboxaldehyde and violacein to protect against fungal infection [17]. A combined analysis of bat skin microbiota and metabolomics further revealed that *Streptomyces buecherae* AC541 isolated from bat skin secreted nigericin, which inhibits *Pd* growth in vitro [18]. Collectively, these studies highlight the intricate interrelationships among microbiota–metabolite–pathogens. However, a systematic understanding of the dynamic changes in bat skin microbiota, metabolites, and their roles in pathogen defense remains lacking.

Building upon this foundation, the study focused on the widely distributed bat species with *Rhinolophus ferrumequinum* in Northeast China. By collecting skin microbial and metabolite samples and integrating 16S rRNA sequencing with non-targeted metabolomic analyses, we investigated the spatiotemporal dynamics of skin microbiota and their metabolites during hibernation. Through correlation analyses, metabolites closely associated with anti-*Pd* microbes were identified and subsequently validated for their inhibitory activity. This study aims to uncover the underlying mechanisms contributing to the notable resistance to *Pd* of *R. ferrumequinum* and its ability to defend against pathogens.

## 2. Materials and Methods

### 2.1. Sample Collection

Epidermal swab and tape samples of *R. ferrumequinum* were collected from December 2023 to April 2024 at Dalazi Cave, Jilin Province, at approximately 60-day interval during the hibernation period. Sampling time points corresponded to December 2023 (early hibernation), February 2024 (middle hibernation), and April 2024 (late hibernation). In the late hibernation period, additional samples were collected from Temple Cave in Liaoning Province and Gezi Cave in Jilin Province. In total, 49 bats were sampled (detailed information is provided in Appendix A). To detect Pd, sterile polyester swabs moistened with sterile water were used to gently wipe five times along the bats’ muzzle and forearm [19,20]. Using a similar method, the opposite wing membrane (propatagium and plagiopatagium) was swabbed to investigate bacterial community composition. Within 24 h of collection, each swab was stored in a 1.5 mL microcentrifuge tube containing 500 μL of RNAlater^®^ solution (Tiangen, Beijing, China) and kept at −80 °C until DNA extraction. After collecting the skin microbiota samples, three sampling positions were selected on each bat’s wing membrane for metabolite collection using the tape method. D-Squame Standard Sampling Discs (CuDerm, Dallas, TX, USA) were applied with constant pressure for 30 s [21], then carefully removed with sterilized tweezers and stored in cryogenic tubes at −80 °C. The body temperature of each bat was recorded using a Fluke 62 MAX infrared thermometer (Everett, WA, USA). All sampling procedures were completed before the bats fully awoke from torpor. After sampling, we measured each bat’s weight and forearm length, and body mass index (BMI = weight/forearm length) was calculated. All individuals were released back into their natural roosts immediately after sampling. To minimize variability due to differences in sampling technique, all collections were performed by the same researcher. All experimental procedures were approved by the Laboratory Animal Welfare and Ethics Committee of Jilin Agricultural University.

### 2.2. Pseudogymnoascus Destructans Detection

Fungal DNA was extracted from swab samples using a modified DNeasy blood and tissue extraction kit (Qiagen, Hilden, Germany) [19]. The presence of *Pd* was assessed using quantitative PCR (qPCR; Thermo Scientific, Waltham, MA, USA) [22]. The probe sequence was 5′-6(FAM)-CGTTACAGCTTGCTCGGGCTGCC-(BHQ-1)-3′. The primer sequences were as follows: forward primer, 5′-TGCCTCTCCGCCATTAGTG-3′; reverse primer, 5′-ACCACCGGCTCGCTACGTA-3′. The qPCR reaction mixture (20 μL total volume) consisted of 10 μL of 2× QuantiFast Probe PCR Master Mix, 0.25 μL of probe, 5 μL of DNA template, 0.5 μL each of the forward and reverse primers, 2.75 μL of double-distilled water, and 1 μL of ROX Reference Dye. *P. destructans* ATCC MYA-4855 served as the positive control, while ddH_2_O was used as the negative control. Thermal cycling conditions were as follows: initial denaturation at 95 °C for 2 min, followed by 50 cycles of 95 °C for 5 s and 60 °C for 10 s. Each sample was analyzed in duplicate. Based on the qPCR results, bats were classified as infected (Ct value detected) or uninfected (no Ct value). Fungal loads were calculated using the following equation: log_10_(*Pd* [ng]) = (Ct − 22.04942)/3.34789 [19].

### 2.3. Bacterial DNA Extraction and Sequencing

All samples were sequenced by Novogene Co., Ltd. (Beijing, China). The 16S rRNA gene V3–V4 region was amplified using the primers 515F and 806R [23]. Based on the characteristics of the amplified region, a short-fragment paired-end sequencing library was constructed and sequenced on the Illumina NovaSeq 6000 platform (Illumina, San Diego, CA, USA). Sequence processing and analysis were conducted using Quantitative Insights Into Microbial Ecology 2 (QIIME2) [24]. The DADA2 algorithm was employed for quality control and denoising to generate amplicon sequence variants (ASVs) [25]. Across different hibernation periods, a total of 2,929,712 raw reads were obtained (average of 104,633 reads per sample; range: 75,694–118,674), with an average read length of 420 bp. Across different sampling sites, a total of 2,997,208 raw reads were obtained (average of 103,352 reads per sample; range: 70,387–118,674), with an average read length of 419 bp. In total, 44,772 ASVs were identified. Taxonomic classification of each ASV was performed using a pre-trained Naïve Bayes classifier implemented in the classify-sklearn algorithm of QIIME2 (Version QIIME2-202202), based on the SILVA 138.1 reference database [26,27].

### 2.4. Untargeted Metabolomics Sequencing

Untargeted metabolomics sequencing was conducted at Novogene Co., Ltd. (Beijing, China). A total of 100 mg of tissue sample, ground in liquid nitrogen, was transferred to an EP tube, followed by the addition of 500 μL of 80% methanol aqueous solution. The mixture was vortexed, incubated on ice for 5 min, and centrifuged at 15,000× *g* and 4 °C for 20 min. An aliquot of the resulting supernatant was diluted with mass spectrometry-grade water to achieve a final methanol concentration of 53%, followed by another centrifugation under the same conditions. The supernatant was collected for metabolic analysis using ultra-high-performance liquid chromatography (Vanquish UHPLC, Thermo Fisher, Bremen, Germany) coupled with mass spectrometry (Q Exactive™ HF, Thermo Fisher, Bremen, Germany) in both positive and negative ion modes. Molecular feature peaks were identified by matching against the high-quality mzCloud database constructed from standard compounds, combined with the mzVault and MassList databases. Background ions were removed using blank samples. Raw quantitative data were normalized to obtain relative peak areas, and compounds with a coefficient of variation (CV) greater than 30% in QC samples were excluded. This process yielded the final metabolite identification and relative quantification results. Data processing was performed on a Linux operating system (CentOS version 6.6) using R (version 3.4.3) and Python (version 3.5.0). Identified metabolites were annotated using the KEGG database (https://www.genome.jp/kegg/pathway.html, accessed on 22 April 2024), HMDB database (https://hmdb.ca/metabolites, accessed on 21 April 2024), and LIPIDMaps database (http://www.lipidmaps.org/, accessed on 22 April 2024). Based on database annotations, unknown metabolic features were removed, resulting in a total of 671 different kinds of metabolites available for downstream analyses [28].

### 2.5. Microbial Data Analysis

The *Pd* loads on bat skin were compared across different time points and locations using the Kruskal–Wallis test and one-way analysis of variance (ANOVA). Bacterial community composition on bat skin was assessed at the genus level, considering only taxa with relative abundances greater than 1%. Alpha diversity indices, including the Shannon diversity and Observed richness indices, were calculated to evaluate differences in microbial diversity across spatial and temporal gradients. One-way ANOVA was used to test the statistical significance of these differences. Beta diversity was calculated based on the Bray–Curtis dissimilarity matrix, and non-metric multidimensional scaling (NMDS) plots were generated using the *phyloseq* package in R (v4.4.2) [29]. Permutational multivariate analysis of variance (PERMANOVA) was performed using the *adonis* function in the *vegan* package to assess differences in community structure.

To quantify the relative contributions of ecological processes to bacterial community assembly across different spatiotemporal scales, a null model analysis was performed to partition the effects of drift, selection, and dispersal [30]. The beta Nearest Taxon Index (βNTI) was calculated using the *picante* package based on phylogenetic trees and the relative abundances of ASVs [31]. A value of |βNTI| ≥ 2 indicated that deterministic processes are playing a dominant role in shaping microbial communities, whereas |βNTI| < 2 suggested a stronger influence of stochastic processes. βNTI was combined with the Bray–Curtis–based Raup–Crick index (RCI) to infer the major ecological processes shaping bacterial community structure: homogeneous selection (βNTI < −2), heterogeneous selection (βNTI > 2), homogenizing dispersal (RCI < 0.95 and |βNTI| < 2), dispersal limitation (RCI > 0.95 and |βNTI| < 2), and drift (|RCI| < 0.95 and |βNTI| < 2) [32]. Co-occurrence networks of bacterial taxa were constructed for different hibernation stages using *WGCNA* package based on Spearman correlations [33]. To minimize noise and false-positive associations, only genera present in at least 30% of samples were retained for network construction. Correlations with *p* > 0.001 or |r| < 0.6 were excluded. Network topological properties were estimated using the *igraph* package [34]. Edges appearing only within one subnetwork were defined as specialist edges, whereas those shared among multiple subnetworks were defined as generalist edges. Keystone taxa were identified as highly connected taxa exerting significant influence on community structure and function.

To identify ASVs contributing most to spatial variation in bacterial community composition during the late hibernation stage, indicator species analysis was conducted for ASVs with genus-level relative abundance larger than 2%. The indicator value (*IndVal*) was used to evaluate ASV distribution patterns among predefined groups, ranging from 0 (even distribution across groups) to 1 (exclusive presence in one group). *IndVal* calculations were performed using the *multipatt* function in the *indicspecies* package [35], and statistical significance was assessed with 9999 permutations. *p*-values were corrected for multiple comparisons using the false discovery rate (FDR) method. ASVs with adjusted *p* < 0.05 and *IndVal* ≥ 0.4 were retained as indicator species [36]. The *IndVal.g* parameter was applied to account for unequal sample sizes among groups. Functional prediction of the bat skin bacterial community was performed using PICRUSt2, and linear discriminant analysis effect size (LEfSe) was employed to identify differences in KEGG metabolic pathways across sampling sites [37]. All analyses and visualizations were performed using R version 4.4.2.

### 2.6. Metabolomic Data Analysis

Principal component analysis (PCA) and orthogonal partial least squares discriminant analysis (OPLS-DA) were performed using the Metware Cloud platform (https://cloud.metware.cn/#/home, accessed on 25 August 2025) and the Lianchuan OmicStudio platform (https://www.omicstudio.cn/tool, accessed on 25 August 2025) to evaluate clustering patterns of bat skin metabolites across different spatial and temporal conditions. The variable importance of projection (VIP) values were calculated to reflect both the loading weights and the amount of variance explained by each component. To elucidate metabolic differences among hibernation stages, pairwise comparisons of metabolites were conducted between adjacent time points: middle hibernation vs. early hibernation, late hibernation vs. middle hibernation, and late hibernation vs. early hibernation. Metabolites were defined as significantly different based on the criteria of VIP > 1.0, *p* < 0.05, and fold change (FC) < 0.67 or >1.5 [38].

Significantly differential metabolites were subjected to pathway enrichment analysis based on the KEGG database to identify the major metabolic pathways involved. Metabolites from different sampling sites were clustered into ten groups using *k*-means clustering, and each cluster was characterized according to metabolite class composition. Spearman’s correlation analysis was subsequently performed between the metabolites within the clusters of interest and *Pd* loads to explore potential associations.

### 2.7. Integrated Analysis of Microbiota and Metabolites

ASVs with a relative abundance greater than 2% at the genus level were selected for Spearman’s correlation analysis with key metabolic pathways across different hibernation stages and with specific metabolite clusters across sampling locations. To further explore the potential microbial origins of metabolites, the MetOrigin 2.0 platform was employed for source prediction [39].

### 2.8. Metabolite Validation Experiments

Metabolites for subsequent validation experiments were selected based on the following three criteria: (1) significant negative correlation with *Pd* loads and previously reported antifungal activity; (2) inclusion among the top 30 metabolites with the highest VIP values, exhibiting high connectivity in the integrated microbiome–metabolome analysis and documented antifungal activity; and (3) significant positive correlation with microbiota reported to possess anti-*Pd* effects [40]. According to these screening criteria, nine metabolites (Appendix A) were selected for inhibition assays using 96-well microplates. For each well, 50 μL of *Pd* spores (2 × 10^6^ spores/mL) and 50 μL of a 10-fold diluted metabolite solution were added to the experimental group. The positive control consisted of 50 μL of *Pd* spores and 50 μL of 1% tryptone broth, while the negative control contained 50 μL of heat-inactivated *Pd* spores (incubated in a 60 °C water bath for 45–60 min) and 50 μL of 1% pancreatin broth. All plates were incubated at a constant temperature of 13 °C. The absorbance at 492 nm was measured daily from day 0 to day 7 using a spectrophotometer (Multiskan FC, USA). The inhibition rate was calculated as follows: Inhibition rate (%) = 1 − (OD_experiment_ − OD_negative_)/(OD_positive_ − OD_negative_) × 100% [41].

## 3. Results

### 3.1. Pseudogymnoascus Destructans Infection

For *R. ferrumequinum* at different hibernation stages, *Pd* infection was detected in bats during the middle and late stages of hibernation (Appendix A). There was no significant difference in *Pd* loads between these stages (Kruskal–Wallis test: Chi-squared = 3.57, *p* = 0.059). The mean *Pd* loads were –5.38 and –4.71, respectively, and the *Pd* prevalence rates were 50% and 44.4%. Similarly, there was no significant difference in *Pd* loads among different locations (one-way ANOVA: *F* = 0.124, *p* = 0.884; Appendix A). The mean *Pd* loads in Temple Cave and Gezi Cave were –4.64 and –4.55, respectively. Among these sites, *Pd* prevalence reached 100% in Gezi Cave and 90% in Temple Cave.

### 3.2. Study on the Bacterial Community Structure of Bat Skin in Different Hibernation Stages

#### 3.2.1. Changes in the Bacterial Community Composition of Bat Skin Across Hibernation Stages

At the genus level, the composition of the bat skin bacterial community varied across different stages of hibernation. *Crossiella*, *Nitrospira*, and *Lactobacillus* were predominant during the early stage of hibernation. In the middle stage, *Pseudomonas*, *Arthrobacter*, and *Acinetobacter* were the dominant genera, whereas *Crossiella*, *Arthrobacter*, and *Staphylococcus* predominated in the late stage of hibernation (Figure 1A).

#### 3.2.2. Changes in the Bacterial Community Structure of Bat Skin Across Different Hibernation Stages

The Observed richness (Figure 1B) and Shannon diversity (Figure 1C) of the bat skin bacterial community differed significantly among hibernation stages (Observed richness: *F* = 3.439, *p* = 0.049; Shannon diversity: *F* = 9.519, *p* < 0.001). The overall bacterial community structure also exhibited significant differences among stages (NMDS with stress =0.16; PERMANOVA: Pseudo-*F*_2_,_24_ = 2.73, *p* = 0.001, *R*^2^ = 0.19; Figure 1D).

#### 3.2.3. Drivers of Bacterial Community Assembly on Bat Skin During Hibernation

Null model analysis revealed that the assembly of the bat skin bacterial community was primarily governed by stochastic processes (|βNTI| < 2), which accounted for 92.7% of the total ecological processes (Figure 1E). Among these processes, dispersal limitation was identified as the dominant driver of community assembly, followed by drift and others, as well as homogeneous selection (Figure 1F).

**Figure 1 biology-14-01648-f001:**
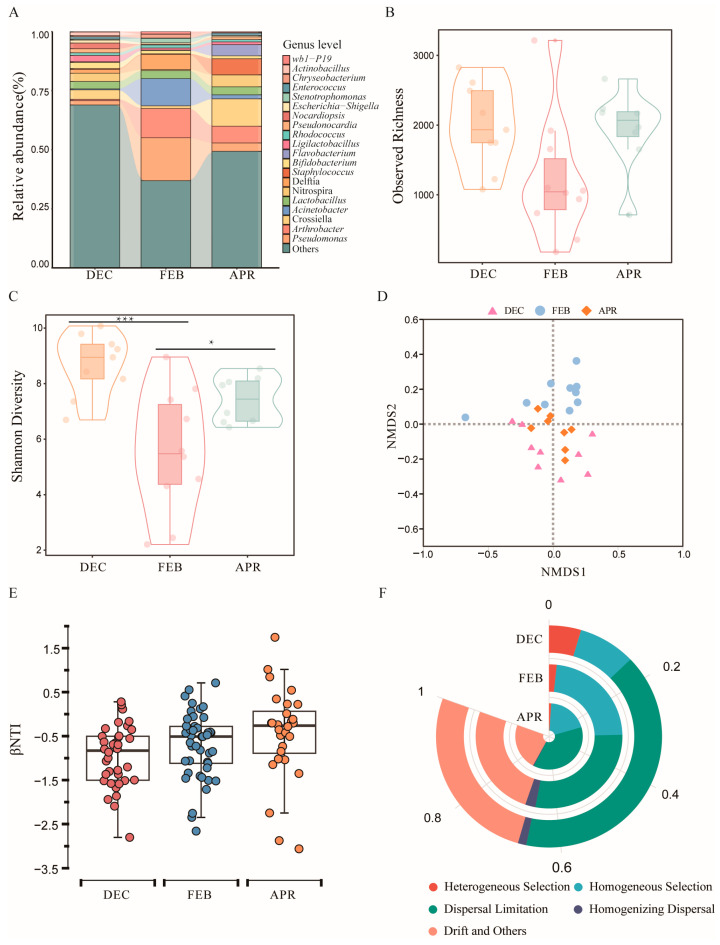
Bacterial community structure of bat skin across different hibernation stages. (**A**) Alluvial diagram showing the major bacterial genera on bat skin with relative abundances greater than 1%. (**B**) Observed richness of the bat skin bacterial community. (**C**) Shannon diversity of bat skin bacteria, where stars indicate significant differences between groups (* *p* < 0.05, *** *p* < 0.001). (**D**) Beta diversity of the bat skin bacterial community based on Bray–Curtis dissimilarity (NMDS plot). (**E**) Relative contributions of deterministic (|βNTI| ≥ 2) and stochastic (|βNTI| < 2) processes to the assembly of bat skin bacterial communities, shown as box plots. (**F**) Relative contributions of different ecological processes driving bacterial community assembly on bat skin.

#### 3.2.4. Co-Occurrence Network Analysis Revealed the Relationships Among Skin Bacterial Communities of Bats at Different Hibernation Stages

The genus-level interactions within the skin bacterial communities of bats during different hibernation stages were integrated to construct microbial co-occurrence networks. The topological properties of these networks varied across the three hibernation stages (Figure 2A). The network from the early hibernation stage exhibited a higher number of vertices (*n* = 397) compared to those from the middle and late stages, whereas the late hibernation network showed a larger number of edges (*n* = 3106) than the other two stages. The clustering coefficient values of the early (0.78) and middle (0.75) hibernation networks were higher than that of the late hibernation network (0.65). Moreover, the middle hibernation network exhibited a lower value of average betweenness (121.0), while the early (6.46) and late (5.28) hibernation networks had higher values of average separation (Figure 2B). These findings indicate that the microbial communities during the early and late hibernation periods exhibited more complex network structures. A relatively high proportion of specialist edges (existing only in this period) was observed in the early and late hibernation networks (ranging from 82.26% to 85.45%), whereas the middle hibernation network contained a larger proportion (29.52%) of generalist edges (present in at least two periods, Figure 2C). This pattern suggests that, with the progression of hibernation, the bat skin bacterial community established numerous new microbial interactions.

To examine the keystone taxa (the highly connected taxa) within the three co-occurrence networks, the top 10 vertices with the highest degree values were shown for each network (28 genera in total, Figure 2D). *Gardnerella*, *Bifidobacterium*, and *Prevotella_9* were identified as the keystone taxa in the early, middle, and late hibernation networks, respectively, while *Limnohabitans* and *Lentilactobacillus* served as keystone genera shared across more than one network. We further analyzed the positive and negative associations of the genera in each network. Our analysis revealed that most genera—including all keystone taxa—had a greater number of positive correlations than negative ones with other taxa. Only a few genera, such as *Rhodococcus* and unidentified*_Gaiellales* displayed more negative than positive associations (Figure 2E).

**Figure 2 biology-14-01648-f002:**
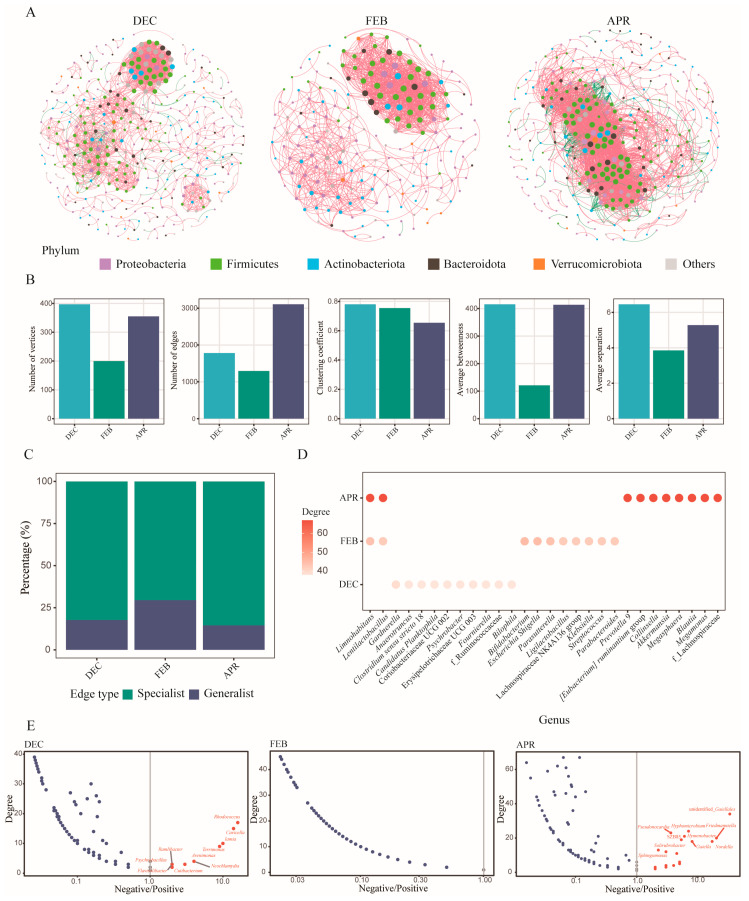
Co-occurrence networks of bat skin bacterial communities during different hibernation stages. (**A**) Phylum-level co-occurrence networks of bat skin bacteria. Red edges indicate positive correlations, while green edges indicate negative correlations. Nodes with the same color belong to the same phylum. (**B**) Topological properties of the bacterial co-occurrence networks. (**C**) Proportions of generalist edges and specialist edges among the three networks. (**D**) Top 10 key taxa in each hibernation stage. The color of each point represents the average degree of the taxon; blank positions indicate taxa that are not identified as a key taxon. (**E**) Scatter plot showing the log-transformed (log10) ratio of negative to positive interactions for each taxon at different hibernation stages. Red nodes represent taxa with more negative than positive interactions, whereas blue nodes represent taxa with more positive than negative interactions.

### 3.3. Changes in the Bacterial Community Structure of Bat Skin Across Different Locations

#### 3.3.1. Variations in the Bacterial Community Composition of Bat Skin Among Locations

At the genus level, distinct differences were observed in the composition of bat skin bacterial communities across the three sampling sites. The bacterial community in Dalazi Cave was dominated by *Crossiella*, *Arthrobacter*, and *Staphylococcus*, whereas *Crossiella*, *Pseudonocardia*, and *Staphylococcus* were predominant in Temple Cave. In contrast, *Staphylococcus*, *Arthrobacter*, and *Rickettsiella* were the dominant genera in Gezi Cave (Appendix A).

#### 3.3.2. Variations in the Bacterial Community Structure of Bat Skin Among Locations

No significant differences were detected in Observed richness (Figure 3A) or Shannon diversity (Figure 3B) among the locations (*p* > 0.05). However, the overall bacterial community structure of bat skin differed significantly among the three caves (NMDS with stress = 0.13; PERMANOVA: Pseudo-*F*_2,25_ = 4.96, *p* = 0.001, *R*^2^ = 0.28; Figure 3C). Indicator species analysis revealed clear spatial variations in the bacterial community structure. *Enterococcus*, *Lactobacillus*, *Ligilactobacillus*, and *Prevotella_9* were significant indicators for Dalazi Cave, whereas *Hyphomicrobium*, *Pseudonocardia*, and *Crossiella* were significant indicators for Temple Cave (Figure 3D).

#### 3.3.3. Driving Forces of Bacterial Community Assembly in Bat Skin Across Different Locations

Null model analysis revealed that the bacterial community assembly in bat skin from Dalazi Cave and Temple Cave was primarily governed by stochastic processes (|βNTI| < 2), which accounted for approximately 74% of the ecological processes. In contrast, the bacterial community in Gezi Cave exhibited |βNTI| > 2 (mean = −1.8), indicating that deterministic processes played a dominant role, contributing to about 60% of the ecological processes (Figure 3E). Among the stochastic processes, dispersal limitation was the main factor shaping the aggregation of bacterial communities in Dalazi Cave. In Temple Cave, homogeneous selection was the primary driving force, followed by dispersal limitation. In Gezi Cave, drift and other undetermined processes together with dispersal limitation were the major factors influencing bacterial community assembly (Figure 3F).

#### 3.3.4. Functional Differences in Bacterial Communities of Bat Skin Among Locations

Functional prediction using PICRUSt2 indicated significant differences in the relative abundance of KEGG-annotated genes among locations (LEfSe, LDA score > 2.5, *p* < 0.05). Most predicted gene functions were related to metabolic pathways. Carbohydrate metabolism represented the largest proportion (33.3%) in Dalazi Cave, whereas amino acid metabolism was dominant in both Temple Cave (26.7%) and Gezi Cave (50%) (Appendix A).

**Figure 3 biology-14-01648-f003:**
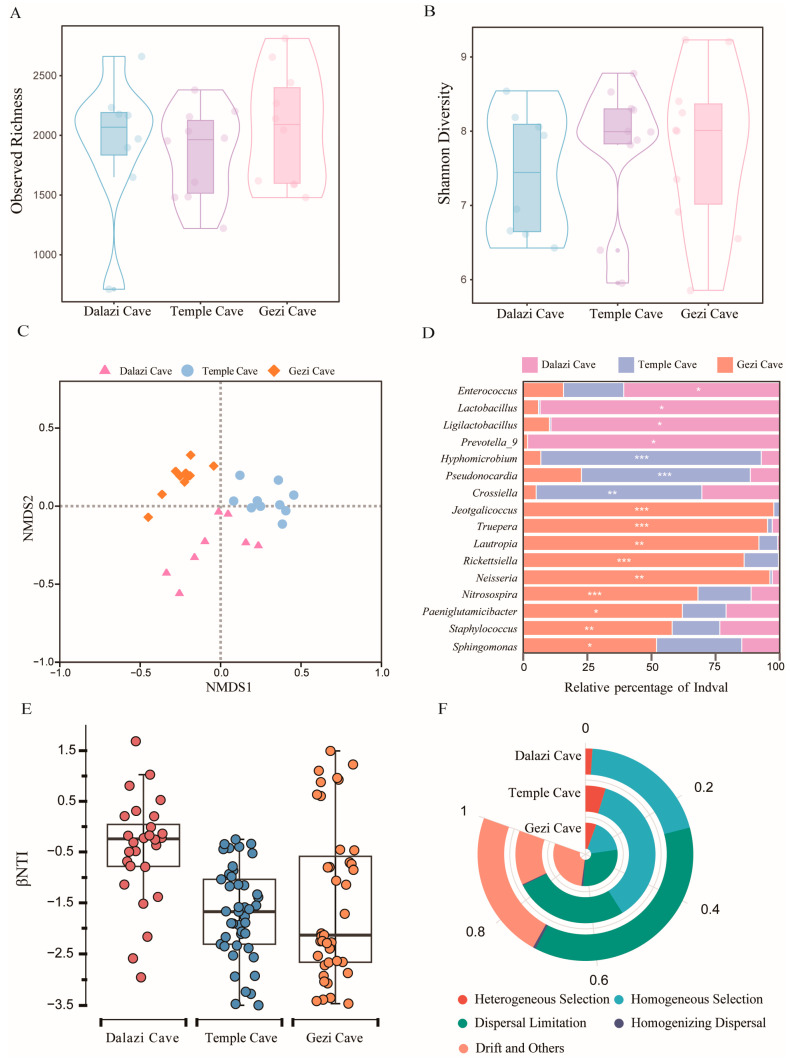
Bacterial community structure of bat skin across different locations. (**A**) Observed richness of bat skin bacteria. (**B**) Shannon diversity index of bat skin bacteria. (**C**) Beta diversity of bat skin bacteria visualized by NMDS plot based on the Bray–Curtis dissimilarity matrix. (**D**) Significant indicator taxa of bat skin bacteria. The analysis was based on ASVs with relative abundance greater than 2%. Statistical significance between groups is indicated by stars (* *p* < 0.05, ** *p* < 0.01, *** *p* < 0.001; *IndVal* ≥ 0.4). (**E**) Relative contributions of deterministic (|βNTI| ≥ 2) and stochastic (|βNTI| < 2) processes to the assembly of bat skin bacterial communities, shown as boxplots. (**F**) Relative contributions of different ecological processes driving the assembly of bat skin bacterial communities.

### 3.4. Study on Skin Metabolites of Bats During Different Hibernation Stages

#### 3.4.1. Changes in Skin Metabolites of Bats Across Different Hibernation Stages

Untargeted metabolomics analysis revealed dynamic changes in the metabolites of bat skin during hibernation. PCA and OPLS-DA demonstrated clear clustering patterns among phases (Figure 4 and Appendix A). To identify metabolites that contributed most to group separation, the top 30 metabolites with the highest VIP scores were selected for visualization. Among them, morphine, perillartine, and gamithromycin were the top three metabolites driving group differentiation (Figure 4B).

#### 3.4.2. Detection of Differential Metabolites in Bat Skin at Different Hibernation Stages

A total of 126 differential metabolites were identified between the middle hibernation and early hibernation stages, 68 metabolites between the late hibernation and middle hibernation stages, and 72 metabolites between the late hibernation and early hibernation stages. Three metabolites were found to be shared among hibernation stages (Figure 4C). During the hibernation process, the concentrations of morphine and perillartine gradually increased, whereas gamma-glutamylcysteine exhibited an opposite trend (Figure 4D). KEGG functional enrichment analysis revealed that the Metabolic pathways (ko01100), Pyrimidine metabolism (ko00240), and Arginine and proline metabolism (ko00330) were significantly enriched (*p* < 0.05) in the middle hibernation and early hibernation stages. Toluene degradation (ko00623) and Aminobenzoate degradation (ko00627) were significantly enriched (*p* < 0.05) in the late hibernation and middle hibernation stages, while the Other carbon fixation pathways (ko00720) and Citrate cycle (TCA cycle, ko00020) were significantly enriched (*p* < 0.05) in the late hibernation and early hibernation stages (Figure 4E).

Moreover, KEGG annotation of up- and down-regulated metabolites indicated that the Linoleic acid metabolism (ko00591) was significantly up-regulated in the late hibernation stage compared with both middle and early hibernation (*p* < 0.05). In contrast, the Glutathione metabolism (ko00480) was significantly down-regulated in middle hibernation compared with early hibernation (*p* < 0.05), and the Arachidonic acid metabolism (ko00590) was significantly up-regulated in late hibernation compared with early hibernation (*p* < 0.05, Appendix A). The metabolites involved in these three pathways were selected for further analysis.

**Figure 4 biology-14-01648-f004:**
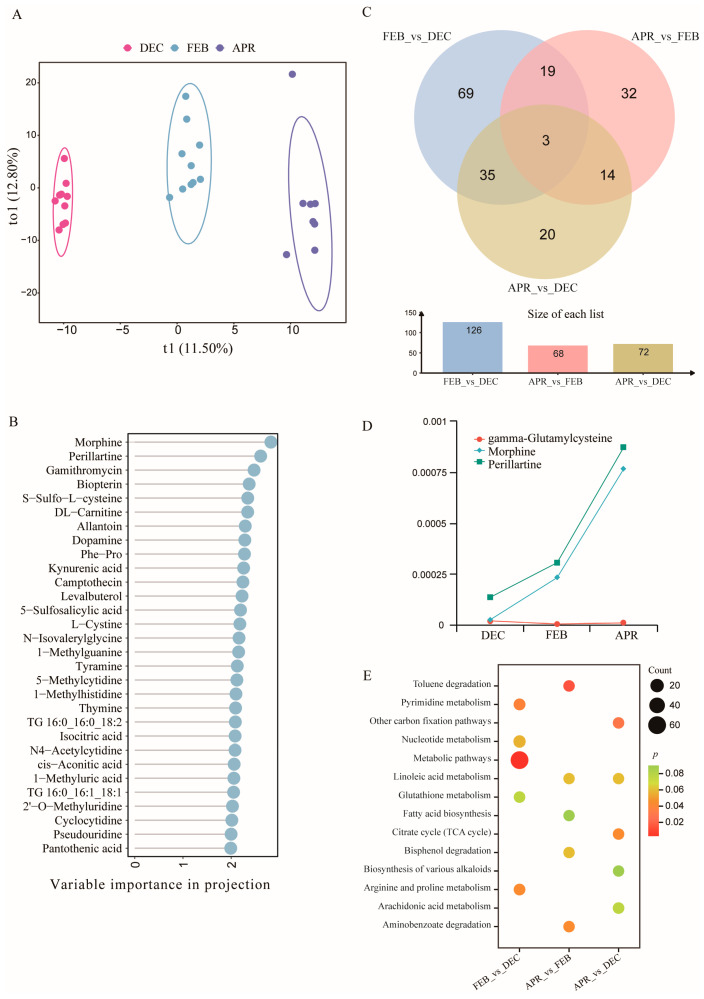
Changes in bat skin metabolites across different hibernation stages. (**A**) OPLS-DA score plot of bat skin metabolites. (**B**) VIP scores ranking metabolites according to their contribution to group separation. (**C**) Venn diagram illustrating the differential metabolites among the three pairwise comparisons: middle hibernation vs. early hibernation, late hibernation vs. middle hibernation, and late hibernation vs. early hibernation. (**D**) Variation patterns of common differential metabolites across the three stages. (**E**) The top five KEGG pathways were significantly enriched in each comparison.

### 3.5. Study on Skin Metabolites of Bats from Different Locations

#### 3.5.1. Variations in Skin Metabolites Among Bats from Different Locations

The metabolic profiles of bat skin samples collected from different locations were analyzed. OPLS-DA revealed a clear separation of metabolites among the groups (Figure 5A), whereas PCA showed no distinct clustering pattern (Appendix A). D-(+)-Maltose, Oxytetracycline, and Phenylpyruvic acid were identified as the top three metabolites contributing to group discrimination (Figure 5B).

#### 3.5.2. Metabolite Clusters of Bat Skin in Different Locations

To elucidate the variation in skin metabolites of bats across different locations, a total of 671 metabolites were grouped into ten clusters, each exhibiting distinct abundance trajectories (Figure 5C). With changes in location, the abundance of metabolites in cluster 8 (62 metabolites) gradually increased, whereas those in cluster 9 (64 metabolites) gradually decreased. The metabolites in cluster 1 (68 metabolites), cluster 3 (64 metabolites), and cluster 5 (51 metabolites) showed an increasing then decreasing trend, while those in cluster 4 (61 metabolites), cluster 6 (43 metabolites), and cluster 7 (89 metabolites) displayed the opposite pattern—first decreasing and then increasing. The metabolites in cluster 2 (121 metabolites) and cluster 10 (48 metabolites) exhibited opposite trends, with abundances, respectively, increasing and decreasing after the Temple Cave (Figure 5E). Overall, the metabolites in the ten clusters were mainly composed of lipids and lipid-like molecules as well as organic acids and their derivatives. In addition, cluster 7 contained a higher proportion of organoheterocyclic compounds, whereas cluster 8 was enriched in benzenoids (Figure 5D).

Spearman correlation analysis was conducted for cluster 8 and cluster 9, which exhibited similar and opposite trends, respectively, in relation to *Pd* loads across different locations. In cluster 8, 2-Amino-1,3,4-octadecanetriol showed a significant negative correlation with *Pd* loads (*p* < 0.05), whereas six metabolites, including N-Acetylvaline, were significantly positively correlated with *Pd* loads (*p* < 0.05). In cluster 9, 2-Phenylglycine, Astaxanthin, and T-2 toxin were significantly negatively correlated with *Pd* loads (*p* < 0.05), while five metabolites, such as Homovanillic acid, were significantly positively correlated (*p* < 0.05, Figure 5F).

**Figure 5 biology-14-01648-f005:**
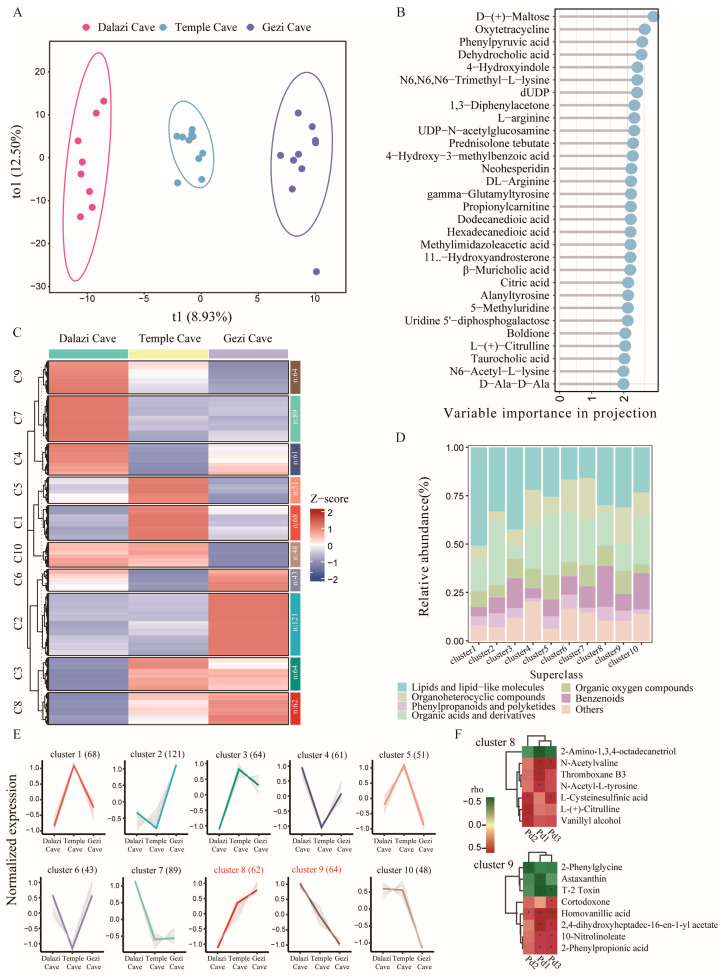
Changes in skin metabolites of bats across different locations. (**A**) OPLS-DA score plot of metabolites in bat skin. (**B**) VIP scores ranking of metabolites. (**C**) Heatmap showing ten metabolite clusters identified by *k*-means clustering. (**D**) Summary of pathway analysis for each cluster. (**E**) Abundance trends of metabolites in ten clusters across different locations. (**F**) Correlation heatmap of cluster 8 and cluster 9 with *Pd* loads. Stars represent significant differences between groups (** *p* < 0.01, * *p* < 0.05).

### 3.6. Joint Analysis of Bat Skin Microbes and Metabolites

#### 3.6.1. Combined Analysis of Bat Skin Microbes and Metabolites During Hibernation

At the genus level, the Spearman algorithm was applied to assess correlations between bacterial taxa with relative abundances greater than 2% and metabolites involved in pathways of interest. The results revealed that 9-Oxo-ODE in the Linoleic acid metabolism (ko00591) was significantly positively correlated with ten genera, including *Lactobacillus* (*p* < 0.05), while (±)12(13)-DiHOME was significantly positively correlated with *Acinetobacter* (*p* < 0.05, Figure 6A). In the Glutathione metabolism (ko00480), both β-nicotinamide adenine dinucleotide phosphate (β-NADP) and L-glutamic acid were significantly negatively correlated with seven genera, respectively (*p* < 0.05). Conversely, (5-L-glutamyl)-L-amino acid exhibited significant positive correlations with seven genera (including *Acinetobacter*, *Arthrobacter*, *Flavobacterium*, and *Pseudomonas*) (*p* < 0.05, Figure 6B). The above four genera were predicted to possess the ability to synthesize (5-L-glutamyl)-L-amino acid (Appendix A). In addition, *Pseudomonas* showed a significant positive correlation with L-glutamic acid, and *Acinetobacter* was positively correlated with L-glutathione oxidized (*p* < 0.05, Figure 6B). Both genera were predicted to be capable of synthesizing L-glutamic acid and L-glutathione oxidized, respectively (Appendix A). In the Arachidonic acid metabolism (ko00590), prostaglandin E2, thromboxane B2, and prostaglandin H2 were significantly positively correlated with seven, ten, and four genera, respectively (*p* < 0.05). *Paenisporosarcina* was identified as the common genus correlated with all three metabolites (Figure 6C).

#### 3.6.2. Combined Analysis of Bat Skin Microbiome and Metabolites Across Different Locations

At the genus level, the Spearman correlation was applied to assess the associations between bacterial genera with a relative abundance greater than 2% and the metabolites in cluster 8 and cluster 9. In cluster 8, *Rickettsiella* exhibited a significant positive correlation (*p* < 0.05) with eleven metabolites, while Neohesperidin showed significant positive correlations (*p* < 0.05) with five genera (Figure 6D). In cluster 9, Saccharin and Melatonin were significantly positively correlated with *Acinetobacter* and *Pseudomonas*, respectively (*p* < 0.05, Figure 6E).

### 3.7. Metabolite Inhibition Experiment

L-Arginine, Astaxanthin, L-Glutamic acid, (5-L-Glutamyl)-L-Amino acid, L-Glutathione oxidized, Prostaglandin E2, Neohesperidin, Saccharin, and Melatonin were selected for *Pd* inhibition assays in vitro. All tested metabolites effectively inhibited *Pd* growth. Among them, Melatonin (94.15%) and Prostaglandin E2 (93.18%) exhibited the strongest inhibitory effects, whereas L-Arginine showed the lowest inhibition rate (48.41%) (Figure 6F).

**Figure 6 biology-14-01648-f006:**
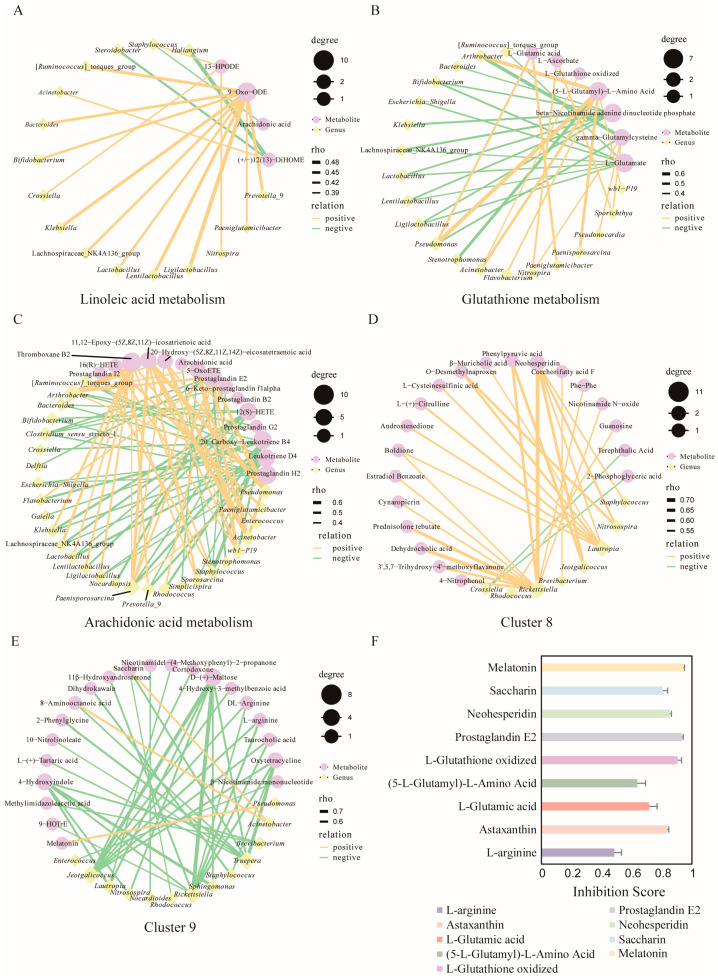
Combined analysis of bat skin microbes and metabolites. (**A**–**C**) Correlation network diagrams of metabolites involved in the Linoleic acid metabolism (**A**), Glutathione metabolism (**B**), and Arachidonic acid metabolism (**C**) during hibernation, showing associations with bacterial genera whose relative abundances exceeded 2% (*p* < 0.05). (**D**,**E**) Correlation network diagrams of metabolites in cluster 8 (**D**) and cluster 9 (**E**) across different locations during late hibernation, showing associations with bacterial species having relative abundances greater than 2% (*p* < 0.05, *R* > 0.5). (**F**) Comparison of *Pd* inhibition rates among nine metabolites. Data are presented as mean ± standard deviation from six independent experiments.

## 4. Discussion

This study systematically investigated the spatiotemporal dynamics of the skin microbiota and metabolites in *R. ferrumequinum* during hibernation, with a focus on exploring potential interactions between these components. By integrating high-throughput sequencing and untargeted metabolomics, we comprehensively characterized the temporal and spatial variations in the skin microbiome and metabolome. These results suggest that potentially beneficial bacteria within the skin microbiota may produce antifungal metabolites that help counteract *Pd* infection, thereby contributing to lower *Pd* burden in *R. ferrumequinum*.

Research has demonstrated that the structure of bat skin microbial communities undergoes significant changes throughout the hibernation period. During middle hibernation, the alpha diversity of the bat skin microbiota was markedly lower than that observed in the early and late hibernation phases (Figure 1B,C), potentially reflecting adaptive adjustments of the skin bacterial community in response to fluctuating winter environmental conditions [6]. Middle hibernation typically coincides with periods of lower ambient temperatures, while the bats’ reduced body temperature (Appendix A) may further constrain the growth of certain bacterial taxa on the skin [42,43]. Moreover, NMDS based on Bray–Curtis dissimilarity revealed distinct clustering patterns among samples (Figure 1D), indicating significant differences in skin microbiome composition across the different hibernation stages. This observation is consistent with previous studies reporting seasonal variations in bat skin bacterial community structures [9].

Understanding the assembly processes of the skin bacterial community is essential for elucidating how the skin microbiome evolves across hibernation phases [44]. The results revealed that stochastic processes—particularly dispersal limitation—serve as the dominant mechanisms regulating the assembly of bat skin bacterial communities (Figure 1E,F), which is consistent with findings on the assembly of bat skin fungal communities [45]. This suggests that bats may acquire diverse bacterial taxa through environmental contact during hibernation. However, dispersal limitation exerts a dual influence on both pathogenic and probiotic bacteria: while it may restrict the transmission of *Pd*, it could also impede the spread of beneficial antagonistic bacteria among populations, thereby influencing the potential efficacy of microbial regulation and disease resistance [9,46,47].

Microbial co-occurrence networks are widely used to investigate relationships within microbial communities, where network interactions reflect varying degrees of microbial competition or niche differentiation. These findings align with observations of microbial networks in other ecosystems [48]. In this study, the co-occurrence networks displayed a higher number of nodes and edges during the early hibernation and late hibernation phases (Figure 2A,B), suggesting greater network complexity. Keystone taxa within such networks play more pivotal roles than other microbiota, and their loss may cause the collapse of modules or even the entire network [49]. *Limnohabitans* and *Lentilactobacillus*, identified as keystone taxa across all time periods (Figure 2D), likely contribute critically to maintaining the stability of the bat skin microbiome. Notably, among the keystone taxa detected during the early hibernation phase, *Psychrobacter* has been reported to exhibit anti-*Pd* activity [50]. Network structures comprising both positive and negative correlations can enhance overall stability [51]. Compared with the early and late hibernation phases, positive correlations—indicative of potential synergistic or mutually beneficial interactions—predominated during middle hibernation (Figure 2E). Most microorganisms do not exist independently, and interspecific interactions within ecological networks reflect essential evolutionary pressures during natural selection, such as the reshaping of microbial relationships through the acquisition of adaptive genes [48].

Across different spatial locations, the alpha diversity of bat skin microbial communities showed no significant variation (Figure 3A,B). However, NMDS based on Bray–Curtis dissimilarity revealed distinct clustering patterns among locations (Figure 3C). The geographic proximity of the three sampling sites during late hibernation may partially explain this pattern. Distance–decay relationships suggest that microbial community similarity between any two sites decreases with increasing geographic distance [52]. Additionally, abiotic factors such as elevation and habitat type may account for the observed microbial variations [6].

Indicator species analysis identified *Enterococcus* and *Paeniglutamicibacter* previously shown to inhibit *Pd* growth in vitro—as being more abundant in the bat skin microbiome (Figure 3D), consistent with earlier reports [6,53]. Stochastic processes were found to be the dominant mechanisms shaping the assembly of bat skin bacterial communities across different locations. In Temple Cave, bacterial community assembly was primarily governed by homogeneous selection, followed by dispersal limitation (Figure 3F), suggesting that the stable environmental conditions in this cave impose strong selective pressures on bacterial community composition. In contrast, dispersal limitation emerged as the main driver of bacterial community clustering in Dalazi Cave and Gezi Cave (Figure 3F), consistent with previous findings [54]. Specifically, dispersal limitation occurs when connectivity between distinct habitats is restricted. Over long timescales, periodic shifts in bacterial community composition driven by environmental filtering—encompassing both biotic and abiotic factors—constitute the key processes shaping these communities. Conversely, over shorter timescales, the influence of environmental filtering weakens, allowing stochastic processes such as homogenizing dispersal to exert a greater effect.

To better evaluate interactions between bat skin microbiota and pathogens, the potential functional profiles of skin microbes were analyzed, revealing that most genes were associated with metabolic pathways. Microbial metabolism is a fundamental biological process, particularly amino acid and carbohydrate metabolism, which plays a vital role in maintaining host physiological homeostasis and defense against disease [55,56].

Untargeted metabolomic analysis of bat skin revealed significant alterations in numerous metabolites across different hibernation stages. PCA and OPLS-DA showed distinct clustering of metabolic profiles among samples from different stages of hibernation (Appendix A, Figure 4A), suggesting a systematic remodeling of skin metabolism throughout the hibernation period. Among the top thirty metabolites contributing to group separation, morphine and perillartine were consistently identified across all three pairwise comparisons, with their concentrations gradually increasing over the course of hibernation (Figure 4B,D). Morphine has been detected in various mammalian tissues, where it is involved in analgesic, stress response, and immune regulatory processes [57]. Recent studies further indicate that certain fungi are capable of producing morphine-like compounds [58]. Perillartine possesses well-documented antioxidant and anti-inflammatory properties [59]. Its upregulation in bat skin may be related to the accumulation of metabolites or alterations in skin redox balance under changing temperature and humidity conditions [60].

The overall skin metabolic profile exhibited pronounced dynamic changes across different hibernation stages (Figure 4C), reflecting adaptive metabolic adjustments to environmental temperature, energy availability, and oxidative stress [61,62]. Pathway enrichment analysis revealed significant enrichment in pathways such as aminobenzoate degradation, metabolic pathways (Figure 4 and Appendix A). Linoleic acid metabolism, glutathione metabolism, and arachidonic acid metabolism were particularly associated with host antimicrobial defense. Metabolites derived from linoleic acid can act as antimicrobial fatty acids that directly inhibit pathogenic bacterial growth [63]. Downregulation of glutathione metabolism may impair the skin’s antioxidant and detoxification capacities, thereby disrupting microenvironmental homeostasis [64]. Arachidonic acid metabolism generates diverse inflammatory mediators and antimicrobial molecules, and its upregulation may indicate that bat skin strengthens its immune barrier through inflammation-related metabolic activity during middle to late hibernation [65]. Notably, several differential metabolic pathways were closely associated with bacterial metabolism. Aminobenzoate degradation and toluene degradation are canonical bacterial pathways commonly found in genera such as *Pseudomonas* [66] and *Acinetobacter* [67]. These pathways not only enable bacteria to degrade aromatic compounds and environmental toxins but may also yield bioactive metabolites to combat pathogens [68,69]. Glutathione metabolism is also widespread among bacteria, enhancing their antioxidant defenses and tolerance to stress or antibiotics [70]. Moreover, certain symbiotic bacteria can utilize or convert linoleic acid to generate antimicrobial fatty acid derivatives [71]. In summary, bat-skin-associated bacteria may play a potential role in pathogen suppression by modulating metabolic pathways closely linked to antimicrobial defense, including linoleic acid metabolism, glutathione metabolism, and arachidonic acid metabolism. These findings highlight the potential of host–microbiota co-regulated metabolic networks in reinforcing the defense barrier against pathogen invasion.

Analysis of bat skin metabolites across different locations revealed that, although PCA did not show clear clustering among samples (Appendix A), the OPLS-DA model effectively discriminated samples from distinct sites (Figure 5A). This result suggests that, despite partial overlap in metabolomic profiles among caves, bat skin metabolomes retain distinct biochemical signatures that can differentiate populations [72]. Among the metabolites contributing most strongly to group separation, D-(+)-Maltose, Oxytetracycline, and Phenylpyruvic acid were the top three (Figure 5B), indicating their pivotal roles in the site-specific metabolic differentiation of bat skin. D-(+)-Maltose is an intermediate product of glucose metabolism, and elevated carbohydrate metabolism may be linked to local tissue repair, keratinocyte activity, or microbial energy demands [73,74]. The detection of oxytetracycline may reflect active microbial metabolism within the skin microbiome, as certain skin-associated bacteria can synthesize bioactive compounds with antibacterial properties that inhibit the growth of competing microorganisms [75,76]. Phenylpyruvic acid is a central intermediate in phenylalanine metabolism, and accumulating evidence suggests that microbial-derived aromatic amino acid metabolites serve as key modulators of host immune responses [77].

Cluster analysis revealed distinct metabolic variations among different locations (Figure 5C,E). A total of 671 metabolites were classified into ten clusters, with the abundance trajectories of each cluster reflecting potential patterns of metabolic regulation in bat skin across sites. These dynamic regulatory processes may be associated with temperature gradients, humidity fluctuations, and differential fungal infection pressures experienced by bats in various hibernation habitats [14,78,79]. The metabolites in these ten clusters were mainly composed of lipids and lipid-like molecules as well as organic acids and derivatives (Figure 5D). Lipids are fundamental components of the skin’s structure and barrier integrity, and changes in their composition can substantially influence the skin microbiome, barrier permeability, and immune responses [80]. Organic acids and derivatives commonly act as intermediates or end products of energy metabolism, the tricarboxylic acid (TCA) cycle, and amino acid degradation or biosynthesis pathways. In studies on hibernation and cold adaptation, metabolic reprogramming is often accompanied by pronounced alterations in organic acid composition [81]. Correlation analysis between *Pd* infection intensity in bats from different locations and metabolite abundance within two clusters of interest (Figure 5F) revealed several metabolites that were significantly negatively correlated with *Pd* loads, suggesting potential candidate biomarkers indicative of cutaneous fungal burden. Among these, astaxanthin—a well-characterized carotenoid antioxidant—acts by scavenging free radicals, mitigating oxidative stress, and protecting cell membranes in the skin and other tissues [82]. Under conditions of elevated *Pd* infection, increased oxidative or immune stress at the skin surface may lead to enhanced astaxanthin consumption.

Through integrative analysis, we revealed a close association between the skin bacterial community and key metabolic pathways. (5-L-Glutamyl)-L-Amino Acid exhibited significant positive correlations with four bacterial genera previously reported to possess anti-*Pd* activity [53], namely *Acinetobacter*, *Arthrobacter*, *Flavobacterium*, and *Pseudomonas* (Figure 6B). Significant positive correlations were also observed between *Pseudomonas* and L-glutamic acid, as well as between *Acinetobacter* and L-glutathione oxidized (Figure 6B). Notably, both bacterial genera exhibit metabolic capabilities for synthesizing these respective metabolites (Appendix A) [83,84]. These findings suggest that the skin microbiota may directly contribute to the host’s antioxidant defenses and pathogen resistance through their intrinsic metabolic activity.

Furthermore, neohesperidin exhibited significant positive correlations with five genera (Figure 6D) and has been reported to possess antifungal activity [85]. Within cluster 9, saccharin and melatonin displayed significant positive correlations with the bacterial genera *Acinetobacter* and *Pseudomonas*, respectively—both of which exhibit resistance to *Pd* (Figure 6E). Previous studies have confirmed that certain *Pseudomonas* strains are capable of producing melatonin [86], which not only directly suppresses fungal growth and biofilm formation but also enhances host immune defense mechanisms [87,88]. Finally, experimental validation demonstrated that all the aforementioned metabolites exerted inhibitory effects on *Pd* growth to varying degrees (Figure 6F).

## 5. Conclusions

Our findings demonstrate that the bacterial community structure on bat skin varies significantly across different stages of hibernation. Dispersal limitation is identified as the dominant ecological process shaping the assembly of these bacterial communities, with potential synergistic or mutualistic interactions prevailing during middle hibernation. In contrast, no significant spatial variation was observed in bacterial community diversity across locations, although several microbial taxa with known *Pd*-inhibitory activity were identified as indicator species. Metabolomic analyses revealed alterations in pathways closely related to host antimicrobial defense, such as linoleic acid metabolism, underscoring the importance of metabolic networks in countering pathogen invasion. Furthermore, correlation analyses between microorganisms and metabolites uncovered potential associations and confirmed that nine metabolites exhibited inhibitory effects on *Pd* growth in vitro. Collectively, this study provides novel insights into the interplay among bat skin microbiota, metabolites, and pathogens.

## Data Availability

Publicly available datasets were analyzed in this study. The raw microbial sequence data has been deposited in the NCBI Sequence Read Archive under the accession number PRJNA1344560. The raw metabolomics data has been uploaded to Metabolight and the data is available with the accession number MTBLS13137.

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
