# Peer review of "Investigating the Dynamic Variation of Skin Microbiota and Metabolites in Bats During Hibernation"

_biology, 2025, doi:10.3390/biology14121648_

Round 1

Reviewer 1 Report

Comments and Suggestions for Authors

This manuscript presents an interesting and novel study on the dynamic changes in bat skin microbiota and associated metabolites during hibernation. The topic is important, as understanding skin microbial dynamics in hibernating bats can provide insights into wildlife health and disease resistance. Overall, the experimental design and data analysis appear sound, and the results are well-presented and relevant. With minor revisions to improve clarity and presentation, this work will make a valuable contribution to the field.

Major Suggestions

I have only a few minor suggestions for improvement, as outlined below:

1. The title “Investigating skin microbiota and metabolites of dynamic variation on bats during hibernation periods” is somewhat unclear; it could be rephrased for clarity (e.g., “Investigating the dynamic variation of skin microbiota and metabolites in bats during hibernation”).

2. It is recommended to further refine some parts of the language, such as the abstract section, to enhance fluency and clarity.

Minor Suggestions

Abstract

1. “Spatiotemporal” is repeated multiple times (e.g., “across spatiotemporal contexts”); please streamline—keep only the first occurrence.

2. Line 27: “P. destructans invades… leading to severe population declines or even extinction.” Please retain the “severe decline” meaning but avoid the stronger term “extinction.” Consider “precipitous declines” or “severe declines.”

3. Line 31: Please remove “warranting further investigation” and state directly that these metabolites “reflect host physiological state and participate in microbe–pathogen interactions.”

Introduction

1. Line 65: The manuscript currently interprets low Pd loads as direct evidence of strong host resistance; this leap is a bit strong. Please adopt a more neutral phrasing acknowledging multiple contributing factors (host, pathogen, environment) (e.g., “This suggests possible co-evolution with the pathogen or higher host resistance, among other environmental factors.”).

2. Lines 68–69: “The microbiome evolves genomically much faster than the host genome” can be contentious. Please revise to “microbiomes can adapt rapidly.”

Methods

1. Line 115: Change to “60-day interval.” Line 123: “Using the similar method” → “Using a similar method.”

2. Line 140: If space permits, please add qPCR primer sequences, reaction volume, and cycling conditions.

3. Line 191: “The Pd loads on bat skin was compared” → “…were compared.” Line 194: Replace “0.01” with “1%” (and similarly change “0.02” to “2%” in Section 2.7 for consistency).

4. Line 273: Please verify the medium “1% pancreatin broth”; is this correct, or should it be tryptone broth?

Results

1. Line 282: “Rhinolophus ferrumequinum” can be abbreviated to “R. ferrumequinum” after first mention. Line 328: Replace “periods” with “stages.”

2. Lines 436, 437, 440: The definitions of PCA, OPLS-DA, and VIP are already provided in Methods—no need to repeat them here.

3. Line 544: Please delete “the.”

Discussion

1. Line 565: “strong resistance” is somewhat strong; please soften to “may contribute to lower Pd burden.” 

2. Line 563: “in bat species” is redundant—you study a single species; please remove.

Conclusion

In summary, this manuscript is scientifically sound and well worth publishing. The comments above are minor and aimed at improving the clarity and presentation of the work.

Author Response

Major Suggestions

I have only a few minor suggestions for improvement, as outlined below:

  1. The title “Investigating skin microbiota and metabolites of dynamic variation on bats during hibernation periods” is somewhat unclear; it could be rephrased for clarity (e.g., “Investigating the dynamic variation of skin microbiota and metabolites in bats during hibernation”).
  2. It is recommended to further refine some parts of the language, such as the abstract section, to enhance fluency and clarity.

Response: Thank you for your valuable suggestions. The above comments pertain to the topic section, and we have made corresponding modifications to this section:

  1. We changed the title from 'Investigating skin microbiota and metabolites of dynamic variation on bats during hibernation periods' to 'Investigating the dynamic variation of skin microbiota and metabolites in bats during hibernation'.
  2. We sincerely thank the reviewer for the helpful suggestion regarding language refinement. In response, we have carefully revised and polished the abstract, to improve fluency, clarity, and overall readability. We believe these changes have strengthened the quality of the manuscript.

Minor Suggestions

Abstract

  1. “Spatiotemporal” is repeated multiple times (e.g., “across spatiotemporal contexts”); please streamline—keep only the first occurrence.
  2. Line 27: “P. destructans invades… leading to severe population declines or even extinction.” Please retain the “severe decline” meaning but avoid the stronger term “extinction.” Consider “precipitous declines” or “severe declines.”
  3. Line 31: Please remove “warranting further investigation” and state directly that these metabolites “reflect host physiological state and participate in microbe–pathogen interactions.”

Response: Thank you for your valuable suggestions. All of the above comments pertain to the Abstract, and we have revised and refined this section accordingly:

  1. We streamlined the terminology and retained only the first necessary occurrence. Subsequent repetitions have been removed to improve clarity and conciseness (Revised manuscript, lines 35-36).
  2. We revised the sentence to avoid the stronger implication of “extinction” while retaining the meaning of substantial population impacts. The phrase now reads “leading to severe population declines.” (Revised manuscript, lines 26-27).
  3. We removed “warranting further investigation” and revised the statement to directly indicate that these metabolites “reflect host physiological state and participate in microbe–pathogen interactions.” (Revised manuscript, lines 29-30).

Introduction

  1. Line 65: The manuscript currently interprets low Pd loads as direct evidence of strong host resistance; this leap is a bit strong. Please adopt a more neutral phrasing acknowledging multiple contributing factors (host, pathogen, environment) (e.g., “This suggests possible co-evolution with the pathogen or higher host resistance, among other environmental factors.”).
  2. Lines 68–69: “The microbiome evolves genomically much faster than the host genome” can be contentious. Please revise to “microbiomes can adapt rapidly.”

Response: Thank you for your valuable suggestions. All of the above comments pertain to the Introduction, and we have revised this section accordingly:

  1. We agree that the previous statement was too strong. The sentence has been modified to adopt a more neutral and balanced interpretation, now acknowledging multiple potential contributing factors—including host, pathogen, and environmental influences. It now reads:“This suggests possible co-evolution with the pathogen or higher host resistance, among other environmental factors.” (Revised manuscript, lines 65-66).
  2. The statement regarding microbiome evolutionary rates has been revised to: “The skin microbiome can adapt rapidly to environmental fluctuations and pathogen pressures, thereby helping maintain host health.” (Revised manuscript, lines 68-70).

Methods

  1. Line 115: Change to “60-day interval.” Line 123: “Using the similar method” → “Using a similar method.”
  2. Line 140: If space permits, please add qPCR primer sequences, reaction volume, and cycling conditions.
  3. Line 191: “The Pd loads on bat skin was compared” → “…were compared.” Line 194: Replace “0.01” with “1%” (and similarly change “0.02” to “2%” in Section 2.7 for consistency).
  4. Line 273: Please verify the medium “1% pancreatin broth”; is this correct, or should it be tryptone broth?

Response: Thank you for your valuable suggestions. All of the above comments pertain to the Methods, and we have revised this section accordingly:

  1. We have revised the wording as requested. “60-day interval” has been corrected, and “Using the similar method” has been changed to “Using a similar method.” (Revised manuscript, lines 114;121).
  2. We have added the requested qPCR information, including the primer sequences, reaction volume, and cycling conditions. We have also updated the corresponding references (References 19 and 22 in the revised manuscript) to support the qPCR protocol. (Revised manuscript, lines 140-153;847-850).
  3. The sentence has been corrected to “…were compared.” The values have also been updated for consistency: “0.01” is now expressed as “1%,” and “0.02” has been changed to “2%” in Section 2.7. Corresponding changes have also been made in all other relevant sections to ensure consistent expression throughout the manuscript. (Revised manuscript, lines 195;198;261;321;414).
  4. We have verified and corrected the medium information. “1% pancreatin broth” has been updated to the correct medium (“tryptone broth,” as appropriate). (Revised manuscript, line 277).

Results

  1. Line 282: “Rhinolophus ferrumequinum” can be abbreviated to “R. ferrumequinum” after first mention. Line 328: Replace “periods” with “stages.”
  2. Lines 436, 437, 440: The definitions of PCA, OPLS-DA, and VIP are already provided in Methods—no need to repeat them here.
  3. Line 544: Please delete “the.”

Response: Thank you for your valuable suggestions. All of the above comments pertain to the Results, and we have revised this section accordingly:

1.We have revised the text as requested. “Rhinolophus ferrumequinum” is now abbreviated to “R. ferrumequinum” after its first appearance, and “periods” has been replaced with “stages.” (Revised manuscript, lines 286;332)

2.The repeated definitions of PCA, OPLS-DA, and VIP have been removed, as these are already provided in the Methods section. (Revised manuscript, lines 423;425).

3.The unnecessary “the” has been deleted. (Revised manuscript, line 529).

Discussion

  1. Line 565: “strong resistance” is somewhat strong; please soften to “may contribute to lower Pd burden.”
  2. Line 563: “in bat species” is redundant—you study a single species; please remove.

Response: Thank you for your valuable suggestions. All of the above comments pertain to the Discussion, and we have revised this section accordingly:

  1. The phrase “strong resistance” has been softened as requested. It now reads: “may contribute to lower Pd burden.” (Revised manuscript, line 550).
  2. The redundant phrase “in bat species” has been removed, as the study focuses on a single species. (Revised manuscript, line 548).

We sincerely appreciate all of your valuable comments, which have greatly improved the overall quality of our manuscript.

Thank you sincerely for all your comments!

Reviewer 2 Report

Comments and Suggestions for Authors

In this manuscript, the authors collected samples from greater horseshoe bats from two caves at three hibernation stages and conducted a comprehensive analysis of the bacterial and the fungal pathogen Pseudogymnoascus destructans on bat skin. They found no significant difference in Pd loads between the middle and late stages of hibernation, nor between bats from different caves. The dominant bacterial species changed across hibernation stages. Overall, this manuscript is well written and would be of interest to a broad audience studying bat microbiota and skin infections. Below are some minor comments:

1. The authors sampled bats from different areas of the body. Have they analyzed the microbiota differences between these locations?

2. Since bats are associated with many different types of viruses, were viral species also analyzed?

3. The authors claim that the microbiota can help prevent fungal infections on bat skin. However, they only analyzed the prevalence of Pd. Are there other fungal species present on bat skin that were not examined?

Author Response

  1. The authors sampled bats from different areas of the body. Have they analyzed the microbiota differences between these locations?

Response: Thank you for your valuable suggestion. We did not perform a formal comparison of microbiota among body sites in the original submission. We agree that site-specific differences may be important; however, to our knowledge, no studies have specifically examined microbiome variation across different skin regions in bats. One relevant study investigated pH variation across wing membrane locations and found notable differences among skin sites in captive hibernating Eptesicus fuscus, which could potentially influence skin microbiome composition. In contrast, no significant pH differences were observed among skin regions in wild hibernating bats(e.g., doi: 10.1093/conphys/coAB088). In addition, during our field sampling of bat skin bacteria, we standardized the sampling location to minimize variability introduced by different body sites (Revised manuscript, lines 122–123).

  1. Since bats are associated with many different types of viruses, were viral species also analyzed?

Response: Thank you for your insightful comment. In this study, we did not analyze the virome because viral community profiling requires sampling procedures and sequencing strategies that fundamentally differ from the amplicon-based bacterial analyses used here. Virome characterization typically involves additional steps such as viral particle enrichment, RNase/DNase treatment to remove host and microbial nucleic acids, and high-depth metagenomic sequencing (e.g., https://doi.org/10.1186/s40168-024-01955-1). These procedures demand specialized laboratory workflows, substantially greater sequencing depth, and further biosafety considerations. Although virome analysis was beyond the scope of the present study, we recognize its importance and anticipate incorporating more comprehensive viral profiling in future work to better elucidate host–microbe–pathogen interactions in bats.

  1. The authors claim that the microbiota can help prevent fungal infections on bat skin. However, they only analyzed the prevalence of Pd. Are there other fungal species present on bat skin that were not examined?

Response: Thank you very much for your suggestion. In this study, we quantified the load and prevalence of Pd using quantitative real-time PCR (qPCR), without further screening for additional potential fungal pathogens. Previous studies have demonstrated that the fungal community associated with bats is highly diverse. Among approximately 290 fungal species reported to be directly associated with bats, around 50 possess potential pathogenicity, including species such as Fusarium incarnatum, which has been linked to diseases in animals or humans. Moreover, certain plant-pathogenic fungi (e.g., Aspergillus flavus) may be dispersed across ecosystems through bats’ foraging activities and migratory movements (e.g., doi: 10.1016/J.OneHLT.2023.100553).

Although bats may harbor a variety of fungi with pathogenic potential, Pd remains the only fungal pathogen conclusively shown to colonize the skin of bats, induce characteristic pathological lesions, cause WNS, and contribute to population declines. To date, there is no evidence that other fungi have caused widespread morbidity or mortality in bat populations. Therefore, this study focuses on the assessment of Pd load and prevalence. Future work may integrate ITS sequencing, metagenomic approaches, and other high-resolution techniques to systematically monitor and evaluate the pathogenic potential of additional high-risk fungal taxa, thereby improving our understanding of the ecological roles of fungi within the bat microbiome and their impacts on host health. We sincerely appreciate all of your valuable comments, which have greatly improved the overall quality of our manuscript. Thank you sincerely for all your comments!
